# Electric Field Sensing with a Modified SRR for Wireless Telecommunications Dosimetry

Fabian Vazquez [1], Alejandro Villareal [1], Alfredo Rodriguez [2], Rodrigo Martin [1], Sergio Solis-Najera [1] and Oscar Rene Marrufo Melendez [3,*]

1 Departamento de Física, Facultad de Ciencias, Universidad Nacional Autonoma de Mexico, Mexico City 04510, Mexico; jfuv@ciencias.unam.mx (F.V.); alejandro.v@ciencias.unam.mx (A.V.); r.martin@ciencias.unam.mx (R.M.); solisnajera@ciencias.unam.mx (S.S.-N.)

2 Department of Electrical Engineering, Universidad Autonoma Metropolitana Iztapalapa, Mexico City 09340, Mexico; arog@xanum.uam.mx

3 Neuroimage Department, National Institute of Neurology and Neurosurgery MVS, Mexico City 14269, Mexico

* Correspondence: omarrufo@innn.edu.mx

**Abstract:** Split ring resonators (SRRs) have been used extensively in metamaterials, showing a strong localization and enhancement of fields, which significantly improves the sensitivity and resolution of the electromagnetic field sensors. We propose the development of an electric field sensor for 2.4 GHz industrial, scientific, and medical (ISM) band applications, by modifying the renowned SRR to contain three concentric pairs of rings. The reduced size makes the sensor affordable for experiments by inserting it in phantoms in order to measure the specific absorption rate (SAR). Power was transmitted from a patch antenna to SRR, with a varying set of distances 1λ, 2λ, 3λ, or 5λ. Experimental measurements of power were conducted with and without a cylindrical distilled-water phantom with agar (4.54%) and NaCl (0.95%). We then computed the electric and magnetic fields and the SAR using these experimental readings of power for different distances. Our sensor was able to measure power values from 20 nW to 0.3 μW with no phantom, and 1 nW to 10 nW with a phantom, in accordance with the values reported for radiofrequency (RF) dosimetry. The sensitivity as a function of the distance determined for the specific case of a phantom was 0.3 μW/cm.

**Keywords:** split ring resonator; SAR; dosimetry; electric field; sensor





## 1. Introduction

The human population is consistently exposed to natural and human-made sources of nonionizing radiation, such as electromagnetic fields (EMF) used for wireless communications encountered in the frequency range from 300 kHz to 30 GHz, considering the spectrum of 5G technology. Since the introduction of wireless technology, the number of people exposed to EMF, intentionally or unintentionally, has increased dramatically [1]. Furthermore, the telecommunications industry depends heavily on dedicated devices like radiofrequency (RF) antennas to transmit and receive EMF signals. Concerns about the health hazards caused in humans exposed to nonionizing electromagnetic radiation have created a need for probes that can perform accurate measurement of the energy absorbed by biological matter, and the reception frequency range of the sensors should be enhanced due to the use of frequency hopping spread spectrum (FHSS) technology. Dipole-based electric field sensors suffer due to the large size of the wavelength employed in the 2.4 GHz band; reducing the dipole size via loaded antennas has some tradeoffs in the form of lower efficiency and narrower bandwidths [2].

We aimed to develop a metamaterial-inspired sensor able to detect the electric field from frequencies commonly found in the S Band (2-5 GHz). Multiband antennas can be designed with different metamaterials such as split ring resonators (SRR) [3], which

have been experimentally tested for telecommunication applications [4,5]. However, the experimental development of this type of resonator is focused on a specific frequency and a narrow band, or for multiple frequencies. The design presented here is able to sense the average energy generated by a FHSS broadcaster. We understand that this has not been fully investigated for electric field sensing. Moreover, there are few sensors for specific absorption rate (SAR), which are principally based on optical systems; data acquired with optical fibers have been previously reported in [6], but the power levels sent to the phantom are higher than those reported for wireless devices, and the physical measurement of interest is only temperature. On the other hand, the miniaturization of flexible antenna sensors, based on the microstrip patch antenna, has been used in the medical field for monitoring physiological variables [7], but they have not been tested as electric field sensors for dosimetry applications. Recently, the development of sensors inspired by metamaterial structures has, for the optical range and the scalable property of fabrication of some designs [8,9], made possible the employment of optical sensors at lower frequencies, as used for wireless communications. In this paper, we investigated the capabilities and the performance of a modified split ring resonator (SRR) for electric field sensing in the ISM band.

## 2. Materials and Methods

The sensor developed and used in this research is based on the SRR. This structure is made of distributed circuits joined together by two gap capacitances. The layout and dimensions of the principal SRR are shown in Figure 1a. These two split rings are at opposite sides and provide inductances, $L_1$ and $L_2$, and the split gap provides capacitance, $C_1$ and $C_2$, which are depicted as an equivalent circuit in Figure 1b [10]. The splits at the rings and the gap between the inner and outer rings induce the natural frequency of resonance [11]. To determine the optimal number of rings, we used CST Microwave Studio (CST MICROWAVE STUDIO, CST GmbH, Darmstadt, Germany) to perform numerical simulations of how much power is reflected by the sensor structure, and we obtained the $S_{11}$-parameter while varying the number of concentric pairs of rings, $N_{pac}$. Figure 2 shows the simulated $S_{11}$-parameter profiles, varying the number of rings. From this, the best match is the 6 $N_{pac}$ profile, but this is outside of the 2.4 GHz frequency spectrum. The three $N_{pac}$ shows good matching and a wide bandwidth that allows us to measure the ISM band around 2.4 GHz (Figure 3 left).

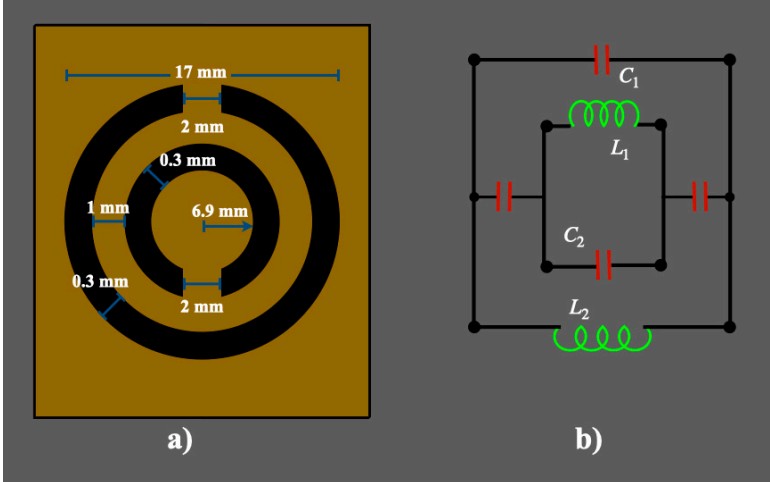

**Figure 1.** Schematic representations of the split ring resonator (SRR) showing dimensions (**a**) and its equivalent circuit (**b**).

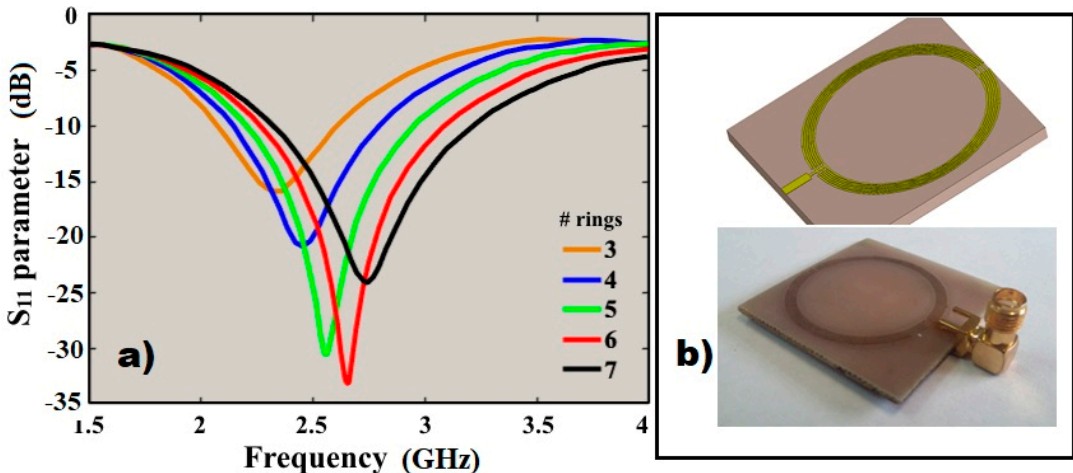

**Figure 2.** (**a**) Simulation of return loss profiles for the metamaterial sensor as a function of the inner SRR number. (**b**) Illustrations of SRR with three rings for simulation and the built design.

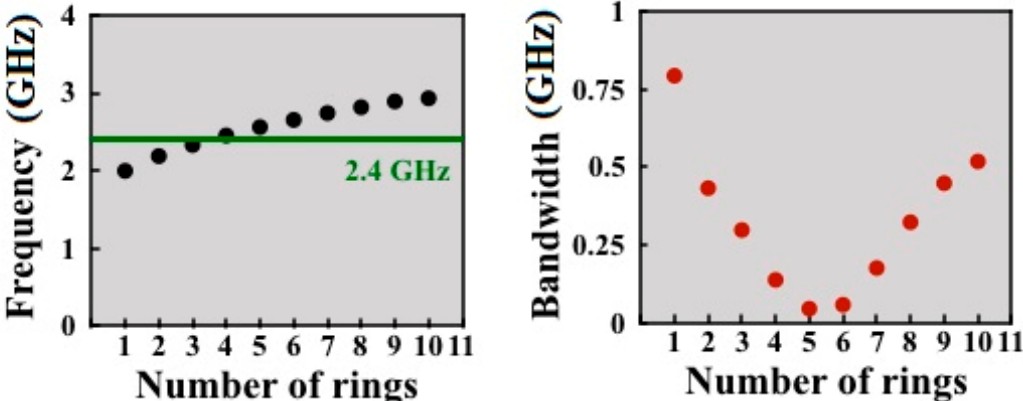

**Figure 3.** (**Left**) Relationship between the concentric SRRs with the bandwidth covered by the E sensor; central frequency is indicated by the 2.4 GHz green line. (**Right**) Bandwidth measured for the tested cases.

We compute the bandwidth for the number of ring pairs, as shown in Figure 3. The four $N_{pac}$ also shows a central frequency close to the 2.4 GHz, but its bandwidth is decreased compared to the three $N_{pac}$.

Consequently, we developed an electric field sensor with three $N_{pac}$. The SRRs were built with two concentric copper loops on a glass-reinforced epoxy laminate material, FR4 PCB ($\epsilon_r = 4.3$); two additional concentric inner ring pairs were added for a total of three $N_{pac}$. The experimental $S_{11}$-parameter is shown in Figure 4 and is compared to the one simulated; both traces are shown, with a reduced span focused in the 2.4 GHz frequency.

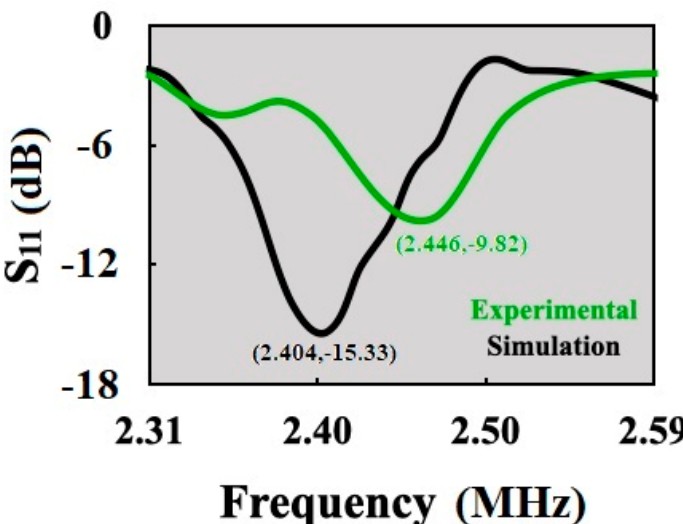

**Figure 4.** $S_{11}$ comparison between simulation (black) and experimentally measured (green) for $3N_{\text{pac}}$.

Additionally, a patch antenna on a glass-reinforced epoxy resin laminate FR4 PCB ($\epsilon_r = 4.5$, tan ($\delta$) = 0.008) with a resonant frequency of 2.44 GHz was constructed to transmit the electromagnetic signals. The effective dimensions were calculated according to [12], so the patch antenna was 2.85 cm $\times$ 3.77 cm and the substrate size was 7.54 cm $\times$ 5.64 cm, with a thickness of 0.14 cm.

To characterize the SRR for electric field sensing, we built a cylindrical (8 cm diameter) phantom filled with agar and a saline solution (NaCl). This cylinder had a volume of 225 cm$^3$ and the solution was composed of distilled water (225 mL), agar (13.5 g), NaCl (2.75 g), and gelatin (5.62 g). This gives a density of 1.04 g/cm$^3$. Experimental power measurements were performed with and without a phantom and with varying distance between the antennas. Figure 5 illustrates the experimental setup for sensing the electric field. The idea is to transmit the controlled RF signal using a patch antenna and SRR for reception purposes. The patched antenna was connected to a RF signal generator (SG6000L DS Instruments, Gardnerville, NV, USA) to generate the controlled RF signal and its power. The SRR was used for RF signal reception and was placed inside a cylinder and connected to a network analyzer (Spectrum Master MS2711D, Anritsu, Kanagawa, Japan) to measure the received power by the resonator. Antennas were separated by a distance that varied according to $n\lambda$, where $n = 2k + 1$, and $k = 0, 1, 2$, and 3. This experiment was repeated using the same distances and antennas, but no cylindrical phantom in order to obtain the energy absorbed by the phantom. The starting power was set to 0 dBm, equivalent to 1 mW at 2.44 GHz, and the initial distance was set to 12.3 cm.

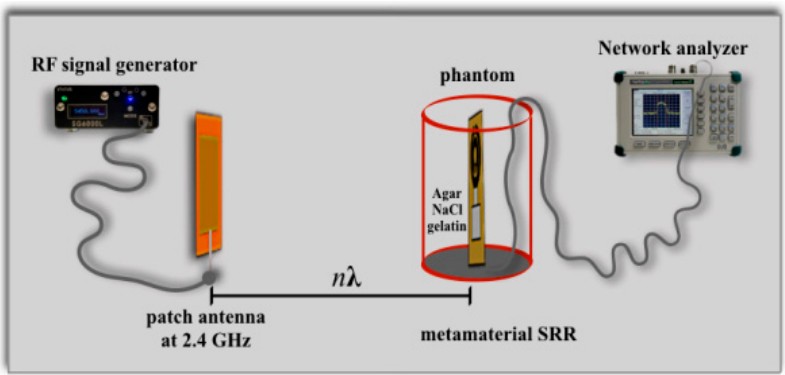

**Figure 5.** Experimental setup showing a patch antenna able to emit electromagnetic energy at different frequencies to a metamaterial SRR within a cylinder filled with a solution to mimic human brain tissue.

### 3. Results

Plots of received power as a function of distance and received power as a function of transmitted power were also computed, as shown in Figure 6. These power readings were acquired at 2.44 GHz and λ = 12.3 cm. As expected, for both cases the dependence of power on separation has a linear pattern, showing a decrease in energy as separation grows. When the phantom is used, a more pronounced decay can be observed since part of the energy is deposited in the phantom. However, both processes have the same pattern and the phantom-related measurements decay more rapidly. To investigate the dependence of the power received from the SRR under various scenarios, linear regressions were obtained with these data (Figure 6).

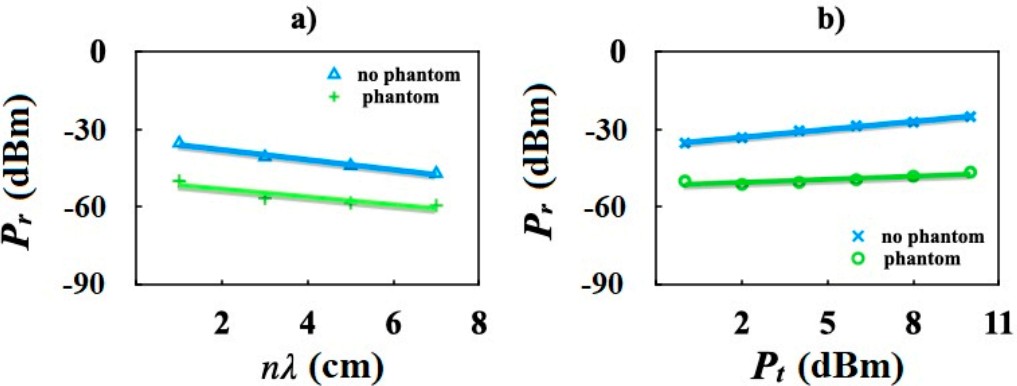

**Figure 6.** Comparison plots of (**a**) distance versus transmitted power with phantom ($P_r$ (phan) = $-1.5080$ ($n\lambda$) $- 50.038$, $R^2$ = 0.9801) and without phantom ($P_r$ = $-1.9175$ ($n\lambda$) $- 34.057$, $R^2$ = 0.8376); and (**b**) transmitted power versus received power with phantom ($P_r$(phan) = 0.3926 $P_t$ $- 51.24$, $R^2$ = 0.7356) and without phantom ($P_r$ = 1.013 $P_t$ $- 35.074$, $R^2$ = 0.9923).

Linear regressions of data in Figure 6 show a similar pattern and linear fits are parallel for this short distance. As expected, as the distance between the two antennas increase the received power decreases (Figure 6a). The linear regression slopes of $P_r$ vs. $n\lambda$ plots show $P_r$(phan) = 0.8 $P_r$, which implies that around 80% of the transmitted power is absorbed by the phantom for this distance interval. These slopes show that the phantom regression has around a 21% lower increase compared to the nonphantom. However, for $P_r$ as a function of the transmitted power, $P_t$, the slopes of the linear fits in Figure 6b show $P_r$(phan) = 0.4 $P_r$, so the energy absorbed by the phantom at 12.3 cm is about half of the absorbed energy without it. The linear fit of the nonphantom case (shown in blue, Figure 6b) may serve as a calibration plot for performing further experiments.

Electric field sensing was conducted via the transmission of power between two antennas, as shown in Figure 5. From these experimental power measurements, we computed the electric field according to the following equation [13]:

$$E_{rms} = \sqrt{\frac{c\mu_0 P_{mean}}{2\pi r^2}}, \tag{1}$$

where $\mu_0$ is the vacuum permeability, $r$ is the separation between antennas, and

$$B_{rms} = \frac{E_{rms}}{c}, \tag{2}$$

where $B_{rms}$ is the magnetic field and $c$ is the speed of light. The exposure to human-made electromagnetic fields can be studied using the concept of SAR [14]:

$$SAR = \frac{\sigma}{2\rho}\left|E_{rms}^2\right| = \frac{\omega\epsilon_0\epsilon''}{2\rho}\left|E_{rms}^2\right| = \frac{\omega c\mu_0\epsilon_0\epsilon''}{4\rho\pi r^2}\left|P_{mean}\right|, \tag{3}$$

where $\sigma$ is the complex conductivity of the solution or tissue, $\omega$ is the resonant frequency, $\epsilon_0$ is the vacuum permittivity, $\epsilon''$ is the imaginary relative permittivity obtained from the dielectric polarization as a result of the interaction of an applied electric field, and $\rho$ is the sample density.

Using the experimental power results of Figure 6 and Equations (1)-(3), we computed both electric and magnetic fields. Figure 7 shows plots of electric and magnetic fields as a function of the distance between them. The ratios of the transmitted and received electric fields as a function of distance were also computed.

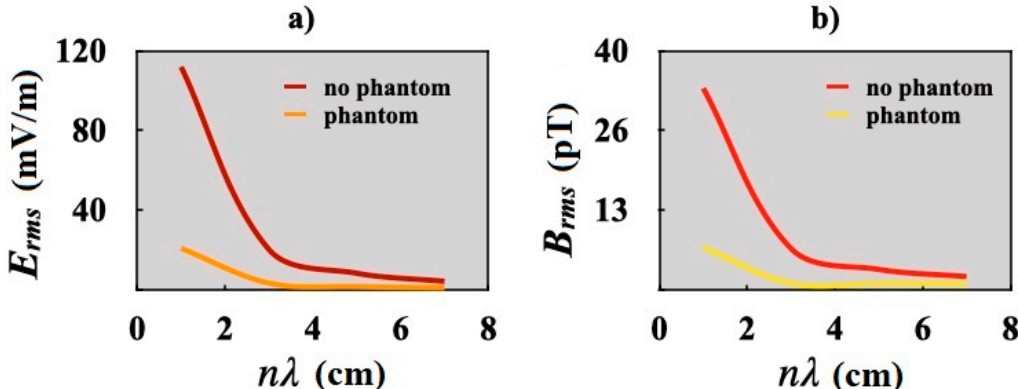

**Figure 7.** Comparison plots of electric fields versus distance (**a**) and magnetic field (**b**) with and without phantom.

## 4. Discussion

Both electric and magnetic fields rapidly decay as a function of the separation of the two antennas, as shown in Figure 7. From Figure 7a,b, we observe the variation of both electric and magnetic fields as the separation between the antennas grows. Most of the energy is in the near field and then decreases rapidly as distance increases away from the antenna. This is clearly a consequence of the power decrease indicated in Figure 6a and Equation (1), and shows an important correlation with the numerically acquired results published by Collardey et al. [15]. The nonphantom electric field results are in accordance with the results reported by Balzano et al. [16], who used thin dipoles with radii of $0.002\lambda$. The energy deposited in the phantom represents around 81% of the electric field energy without a phantom. This indicates that a great deal of energy is absorbed by the phantom, representing an important health hazard to the user of wireless telecommunications.

With the power data and Equation (3), we calculated plots to show the dependence of SAR on the distance, electric field, and applied power (Figure 8).

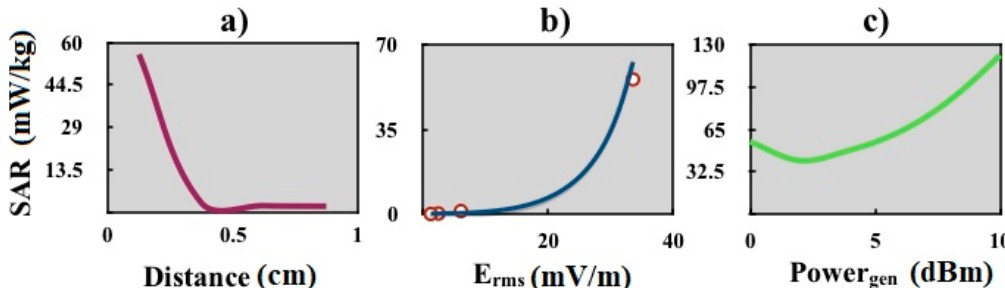

**Figure 8.** Averaged SAR as a function of distance (**a**), electric field (**b**), and applied power (**c**).

The SAR values reported in this research are lower than the limits established for both MRI and telecommunications applications [17,18]. SAR decays rapidly as the separation of antennas increases, and for a distance of $3\lambda$ its value is almost zero. This result is in good agreement with experimental SAR measurements as a function of distance obtained at

1.9 GHZ using a flat phantom and a 3.6 mm diameter antenna [19]. As power is supplied to the SRR, it is expected that the SAR will increase, as shown in Figure 8b.

The range of the measured SAR values are in accordance with the results reported by [20], with a density power capable of producing double DNA (deoxyribonucleic acid) breaks. Nevertheless, the SAR was calculated according to the model reported in [21], which considers the frequency and dielectric properties of a spherical model, but not the distance between the source and the model. Therefore, the computation of SAR accounted for the transmitted power density. Our experimental data consider the received power at a specific point from the source, as it actually occurs for the diverse wireless devices sending energy to humans. Shown by Figure 8, extrapolating to the case $n = 0$ gives values around those reported by [20] and [22] (110 mW/Kg and 180 mW/Kg, respectively), which produces reproductive dysfunction in male mice.

According to [23], the accuracy of the sensor was determined for the design criteria of the central frequency (2.4 GHz) as 1.6% from the experimental frequency achieved. The experimental precision remains below 1% for the criteria of the measured power. These metrics show that our developed sensor is accurate for the intended application.

## 5. Conclusions

By adding concentric pairs of rings to a metamaterial split ring resonator, we show its utility as an electric field sensor capable of measuring the RF energy deposited in human-tissue-mimicking phantoms. Since this device operates within a wide variety of wavelengths, it can be used for a multitude of purposes, from MRI to wireless communications, as an air sensor, or as an insertable sensor in a dielectric medium.

The specific absorption rate results show that the designed sensor can quantify the absorbed power by biological tissues in a more realistic manner. The sensitivity of the sensor allows us to measure expected values for all individuals that are (i) not in direct contact with wireless broadcasters or (ii) are in uncontrolled environments similar to previously reported studies, but exposed to a wide variety of electromagnetic energy transmitters in their daily life.

**Author Contributions:** Conceived and designed the experiments, F.V., S.S.-N., and A.V.; performed the experiments, F.V., S.S.-N., O.R.M.M., and R.M.; analyzed the data, F.V. and A.R.; contributed reagents/materials/analysis tools, F.V., S.S.-N., R.M., A.V., O.R.M.M., and A.R.; wrote the paper, F.V., O.R.M.M., and A.R.; helped to revise the paper, F.V., S.S.-N., R.M., and A.R.; provided financial support, F.V. All authors have read and agreed to the published version of the manuscript.

**Funding:** This work was developed with the support of the program DGAPA-UNAM-PAPIME PE106920.

**Institutional Review Board Statement:** Not applicable.

**Informed Consent Statement:** Not applicable.

**Data Availability Statement:** The data presented in this study are openly available in Dryad at https://doi.org/10.5061/dryad.z612jm68x, reference number.

**Acknowledgments:** We acknowledge Patrick Geeraert for proofreading this manuscript and Laura Aguilar from Instituto Nacional de Rehabilitación Luis Guillermo Ibarra Ibarra for providing a digital repository for the experiments.

**Conflicts of Interest:** The authors declare no conflict of interest.

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
