# Peer review of "Electric Field Sensing with a Modified SRR for Wireless Telecommunications Dosimetry"

_electronics, doi:10.3390/electronics10030295_

Round 1
Reviewer 1 Report
The manuscript by Vazquez and co-authors reports on the electric field sensing by means of SRR for wireless telecommunications dosimetry.
The authors used a multirings split ring resonator to perform some measurements of power at about 2.4GHz.
The topic is rather interesting and the idea is technically sound. Anyway the quality of the presentation (overall) is rather low and the manuscript cannot be accepted in this version.
First of all I recommend a deep english grammar check (better if this is done by a mother tongue expert).
Second, the quality of some figures (nr. 1, 4, 6 and 7) are really bad and not suitable for a peer-review-paper.
Then, the authors mentioned at line 30 "The human population is consistently exposed to natural and man-made sources of nonionizing radiation such as electromagnetic fields (EMF)" this is not correct. also the ionizing radiation (with wavelengths <100nm) is an electromagnetic field!!! maybe the authors want to define the energy range..
At line 39 a reference is probably missing
Line- 46-47: "the design presented here spreads its frequency range to sense the average energy generated for a FHSS broadcaster has not been fully investigated for electric field sensing." this sentence is hard to be understood
Line 60: the authors mention the parameters L _ring and C_gap in the figure 1b...but these are not in the figure. Moreover I cannot understand the rationale behind two different designs to illustrate figure 1a and 1b...
line 64: please define S11-parameter
line 83: please define FR4
the measurements reported in the experimental part are related to a SRR not tuned at 2.4GHz...why the authors did not prepared an optimized SRR? the fabrication seems to be not so complicated.
Fig 8b: please explain the dots in the plot
finally the authors should check also the layout and format used in the document. for example the abstract is written with a different alignment and font.
Reviewer 2 Report
The authors investigated the capabilities and the performance of a modified Split Ring Resonator (SRR) for electric field sensing in the ISM-band. They made an adequate methodology in order to measure the specific absorption rate (SAR).
I have the following suggestions and questions for the authors:
1.- Minor typos
In second page line 46 “; the design presented here spreads its frequency range to sense the average energy generated for a FHSS broadcaster has not been fully investigated for electric field sensing.” It must be revised.
In second page line 48 “ Moreover, There are a...” remove the capital letter from “there”.
2.- I suggest an improve of the introduction, recently there is rapid development in the fabrication of metamaterials and their use as sensors , to improve the manuscript I kindly suggest the following articles to be cited:
- Adv. Optical Mater. 2020, 8, 2000865. https://doi.org/10.1002/adom.202000865
Nanoscale Adv., 2019, 1, 1070-1076 https://doi.org/10.1039/C8NA00250A
3- In the design process (pag.2 line 58) I suggest, if I understood well the design, add to the sentence “The layout and dimensions for the principal SRR are shown in Fig. 1a. “The layout and dimensions for the outer principal pair of SRR are shown in Fig. 1a. “. This clarifies the design geometry. Also, it would be convenient add the distance between pairs of SRRs. I, also, consider could be useful add to figure 1 an inset with the final structure for N_pac=3.
4- How the bandwidth is calculated must be addressed.
5- Usually, permittivity is designed by the Greek letter epsilon instead of mu. I suggest change that for the values of permittivities of FR4 and FR-4 PCB (page 3 line 84 and 87). Also this notation is that I indicate later in the submitted manuscript.
6- In the explanation of equation (3) the authors write, “?′′ is the real relative permittivity”. In this case, due that we are measuring the absorption ?′′ is the imaginary part of the relative permittivity. Please check this part, I suppose it is a typo in the manuscript and the calculation is correct considering the imaginary part. However, the imaginary part of the solution of the phantom is not written in the text. Could the authors clarified this point? This issue must be addressed.
Round 2
Reviewer 1 Report
In the revisited version of the manuscript the authors improved the overall quality.
I recommend the publication